# Partial Identification of Treatment Effects with Implicit Generative Models

**Vahid Balazadeh**[1]    **Vasilis Syrgkanis**[2]    **Rahul G. Krishnan**[1]
[1]University of Toronto, Vector Institute    [2]Stanford University
{vahid, rahulgk}@cs.toronto.edu
vsyrgk@stanford.edu

## Abstract

We consider the problem of partial identification, the estimation of bounds on the treatment effects from observational data. Although studied using discrete treatment variables or in specific causal graphs (e.g., instrumental variables), partial identification has been recently explored using tools from deep generative modeling. We propose a new method for partial identification of average treatment effects (ATEs) in general causal graphs using implicit generative models comprising continuous and discrete random variables. Since ATE with continuous treatment is generally non-regular, we leverage the partial derivatives of response functions to define a regular approximation of ATE, a quantity we call *uniform average treatment derivative* (UATD). We prove that our algorithm converges to tight bounds on ATE in linear structural causal models (SCMs). For nonlinear SCMs, we empirically show that using UATD leads to tighter and more stable bounds than methods that directly optimize the ATE. [1]

## 1   Introduction

Estimating average treatment effects (ATEs) is a common task that arises in fields involving decision-making, such as healthcare and economics. In the presence of the gold-standard randomized controlled trial (RCT) data, one can compare the outcome variable between treated and control groups to make decisions. But RCTs can be costly to set up and run and are, in many circumstances, infeasible. Consequently, communities are using observational data to assist in decision-making.

Identification of treatment effects from observational data is tied to the structure of the causal graph. For example, the treatment $T$ and outcome $Y$ in Figure 1b are confounded by an unobserved random variable, making it impossible to find the causal effect of $T$ on $Y$ only from observational data. On the other hand, Figure 1c is identifiable, and one can adjust for confounders using the Back-door formula [Pearl, 2009]. Even in identifiable settings, non-parametric estimations such as Back-door adjustment formula can point-identify the ATE only with additional assumptions such as positivity, i.e., $P(T = t|X) > 0$ for all values of covariate $X$. Observational data is finite, high-dimensional, and consequently can suffer from severe violations of such assumptions [D'Amour et al., 2021].

In lieu of the challenges of point-identification, there has been a recognition that decisions can be justified using reliable bounds on the ATE rather than its exact value. For an oncologist treating a cancer patient, knowing that a drug has a significant, positive reduction in the patient's risk of progression may suffice as a rationale to prescribe that drug. This problem is known as partial identification [Manski, 2003]. Most existing methods for bounding the ATEs are only applicable in discrete/binary treatment variables [Makar et al., 2020, Zhang et al., 2021, Duarte et al., 2021, Guo et al., 2022]. There has been recent interest in continuous treatment settings. However, such

---

[1]Our code is accessible at https://github.com/rgklab/partial_identification

36th Conference on Neural Information Processing Systems (NeurIPS 2022).

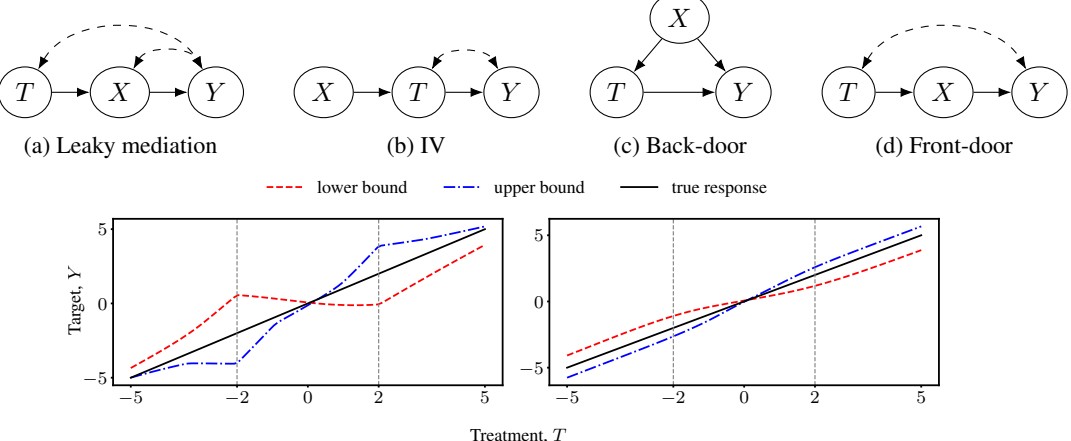

(e) Partial identification of ATE in a finite linear Back-door dataset.

Figure 1: The causal graphs for non-identifiable (a) Leaky mediation and (b) Instrumental Variable (IV), and identifiable (c) Back-door and (d) Front-door settings. $T$, $X$, and $Y$ represent treatment, covariates, and target variables. The dashed double-arrows represent latent factors. (e) The response functions corresponding to partial identification of $\mathbb{E}[Y_{T=2}] - \mathbb{E}[Y_{T=-2}]$ in a Back-door linear SCM after training a generative model to match the distribution. (Left) shows the results for directly optimizing the ATE, which leads to a non-informative bound due to the irregularity of ATE with continuous treatments. (Right) is our solution by optimizing the UATD, which results in a tighter bound. Each point $(t, y)$ represents the expected outcome $y$ after intervention $T = t$ in the learned generative model.

methods are applicable for special causal graphs such as the instrumental variables (IV) setting [Gunsilius, 2020, Kilbertus et al., 2020] or make parametric assumptions on the family of treatment-response functions [Padh et al., 2022]. An exception is the work by Hu et al. [2021] which provides a non-parametric approach for partial identification using generative adversarial networks (GANs). However, they only provide convergence guarantees for the special case of IV causal graphs.

Using the framework of structural causal models (SCMs) and causal graphs, one can see partial identification as a constrained optimization problem, where the objective, i.e., maximizing/minimizing the ATE, can be written as a post-intervention function of exogenous noise (a.k.a response function) and the constraint is to match the generated samples with the observational distribution. This naturally leads to using generative neural networks such as neural causal models (NCMs) [Xia et al., 2021]. We find that directly solving the ATE optimization using flexible generative models such as GANs can lead to non-informative and degenerate solutions. The flexibility afforded by generative models such as GANs allows them to deviate significantly from the true response curve in the neighborhood of intervention points to maximize/minimize the ATE while continuing to generate samples akin to the data distribution. Figure 1e (left) showcases a typical solution to the ATE optimization.

Our insight is that the ATE between any two points can be approximated as an integral over the derivatives of the response function w.r.t. the treatment variable. Rather than directly optimizing the ATE, we optimize the partial derivatives of the response function, a quantity that we refer to as the uniform average treatment derivative (UATD). [2] By optimizing the UATD, the model is required to maximize/minimize the partial derivatives for all points within the treatment support, avoiding extreme local solutions as shown in Figure 1e (right). Our contributions are as follows:

- We formally define the partial identification of average treatment effects as a distributionally-constrained optimization problem, where we choose Wasserstein distance as our constraint metric.
- For the class of linear SCMs, we prove that the solution to our optimization problem converges to optimal bounds on the true value of ATE in infinite data for general causal graphs.
- We use the solution to partial identification of UATDs to find informative bounds on the value of ATE. We introduce a practical algorithm to solve the distributionally-constrained optimization problem using the Lagrange multiplier formulation with alternating optimization. We empirically

---

[2] Average treatment derivative is also known as average partial effect in the literature [Powell et al., 1989, Wooldridge, 2005, Rothenhäusler and Yu, 2019].

show that our algorithm results in tighter and more stable bounds than methods that directly optimize the ATE.

## 2 Problem Setup & Background

We introduce the definitions and assumptions we will use throughout the paper. Consider the observed data as (possibly continuous) random variables $\mathbf{V} = \{X_1, \cdots, X_m, T, Y\} \in \mathbb{R}^d$, where $T$, $Y$, and $\{X_1, \cdots, X_m\}$ denote the treatment variable, target variable, and covariates, respectively.

**Data generating model.** Our approach will be based on the framework of Structural Causal Models (SCMs). An SCM is a tuple $\mathcal{M} = (\mathbf{V}, \mathbf{U}, \mathcal{F}, P_{\mathbf{U}})$, where each observed variable $V_i \in \mathbf{V}$ is a deterministic function of a subset of variables $\mathbf{pa}(V_i) \subseteq \mathbf{V}$ and latent variables $\mathbf{U}_{V_i} \subseteq \mathbf{U}$, i.e.,

$$V_i = f_{V_i}(\mathbf{pa}(V_i), \mathbf{U}_{V_i}) \text{ where } f_{V_i} \in \mathcal{F}, \ V_i \notin \mathbf{pa}(V_i) \tag{1}$$

The only source of randomness are latent variables $\mathbf{U}$ with probability space $(\Omega, \Sigma, P_{\mathbf{U}})$. This induces a probability law over the observed variables $P_{\mathcal{M}}$. We may omit the subscript $\mathcal{M}$ and denote the observational probability distribution by $P$ throughout the text. Given $\mathcal{M}$, one can construct a graph with nodes $\mathbf{V} \cup \mathbf{U}$ and directed edges from nodes in $\mathbf{pa}(V_i) \cup \mathbf{U}_{V_i}$ to $V_i$. We call this graph the causal graph corresponding to SCM $\mathcal{M}$ and denote it by $\mathcal{G}_{\mathcal{M}}$ or simply $\mathcal{G}$. We assume $\mathcal{G}$ is acyclic and known. Moreover, We will assume each node in the graph is 1-dimensional. For random variable $V$ in the SCM $\mathcal{M}$, let $V_{\mathcal{M}}(\mathbf{u})$ be its deterministic value after fixing a realization $\mathbf{u}$ of latent variables $\mathbf{U}$. The causal effect of treatment $T$ on target $Y$ is:

**Definition 1** (Causal Effect). *Let $Y_{\mathcal{M}(T=t)}(\mathbf{u})$ be the value of $Y$ by fixing $\mathbf{U} = \mathbf{u}$ and changing function $f_T$ to a constant function $f_T = t$ in $\mathcal{M}$. Then, we call the random variable $Y_{\mathcal{M}(T=t)}$ the causal effect of treatment $T = t$ on target $Y$. We may simplify the notation and write it as $Y_t$ if the SCM $\mathcal{M}$ and treatment variable $T$ are known from context. Note that $Y_{T(\mathbf{u})}(\mathbf{u}) = Y(\mathbf{u})$.*

When $T$ is continuous, then we can view $\{Y_t : t \in supp(T)\}$ as a stochastic function defined on $(\Omega, \Sigma, P_{\mathbf{U}})$. This is referred to as the response function, partial dependence plot, and dose-response curve in the literature [Zhao and Hastie, 2021, Ritz et al., 2015, Chernozhukov et al., 2018].

**Average treatment effect, average treatment derivative, and partial identification.** Our goal is to estimate bounds on the effectiveness of a treatment regime on a population from the observational distribution $P$ and the causal graph $\mathcal{G}$. In the continuous treatment case, where there is no "on"/"off" notion of treatment, we can compare the average causal effect of an arbitrary treatment (dosage) to the average causal effect at a fixed point $T = t_0$. For example, to indicate the effect relative to not prescribing any treatment, we can choose $t_0 = 0$. This quantity is known as the average treatment effect, average level effect, or average dose effect in the literature on continuous treatment setting [Hirano and Imbens, 2004, Kennedy et al., 2017, Callaway et al., 2021].

**Definition 2** (Average Treatment Effect). *For SCM $\mathcal{M}$, the average treatment effect (ATE) at $T = d$ w.r.t. a fixed point $T = t_0$ is defined as*

$$ATE_{\mathcal{M}}(d) := \mathbb{E}_{\mathbf{u} \sim P_{\mathbf{U}}}[Y_{\mathcal{M}(T=d)}(\mathbf{u}) - Y_{\mathcal{M}(T=t_0)}(\mathbf{u})] \tag{2}$$

Note that estimating the ATE and finding bounds on it only depends on the value of the average response function in $T = d$ and $T = t_0$. As pointed in Gunsilius [2020], this quantity can take arbitrary values if we do not make any assumptions on the set of response functions. Here, we assume the partial derivative of the response function w.r.t. the treatment, i.e., $\partial Y_t / \partial t$ exists and is a bounded continuous function. We then define the average treatment derivative as the following:

**Definition 3** (Average Treatment Derivative). *For the treatment regime $f_T$ in SCM $\mathcal{M}$, we define the average treatment derivative (ATD) as*

$$ATD_{\mathcal{M}} = \mathbb{E}_{\mathbf{u} \sim P_{\mathbf{U}}}\left[\frac{\partial Y_{\mathcal{M}(T=t)}(\mathbf{u})}{\partial t}\bigg|_{t=T(u)}\right], \tag{3}$$

Estimating the ATD can be seen as a proxy for the effectiveness of the prescribed treatment, where we consider the population-level average effect of an infinitesimal increase in the treatment/dosage [Rothenhäusler and Yu, 2019]. In this work, however, we leverage the regularity of this quantity to achieve smoother solutions to the ATE estimation. We will expand on this in section 4.

Note that we cannot readily use eq. 2 (or eq. 3) to estimate the ATE (or ATD), as we only have access to the observational distribution $P$ and the causal graph $\mathcal{G}$ and not the latent distribution $P_{\mathbf{U}}$. In fact, ATEs are generally non-identifiable, i.e., there exist multiple SCMs with the same causal graph $\mathcal{G}$ and generated distribution $P$ that result in different values of ATE. For some graphs, however, one can use non-parametric identification algorithms like $do$-calculus to identify the causal effect from the observational distribution [Pearl, 2009]. In practice, even for identifiable causal graphs, we cannot pinpoint the true ATE due to the uncertainty caused by sampling variation and finite sample errors. Instead, we are interested in finding a tight set of possible solutions that will contain the true value of ATE (or ATD) with high probability. This is known as the partial identification problem in the literature [Manski, 2003].[3] More formally, the partial identification of ATDs/ATEs is defined as:

**Definition 4** (Partial Identification of ATD/ATE). *Partial identification of ATD is the solution to the following optimization problem:*

$$\left( \min_{\mathcal{M}' \in \mathfrak{M}} ATD_{\mathcal{M}'}, \max_{\mathcal{M}' \in \mathfrak{M}} ATD_{\mathcal{M}'} \right) \text{ s.t. } P_{\mathcal{M}'} = P \ \& \ \mathcal{G}_{\mathcal{M}'} = \mathcal{G} \tag{4}$$

*where $\mathfrak{M}$ is the set of all SCMs on random variables $\mathbf{V}$. We denote the solution to the above problem as $(\underline{ATD}, \overline{ATD})$. Similarly, we can define the partial identification of ATEs by replacing $ATD_{\mathcal{M}'}$ with $ATE_{\mathcal{M}'}(d)$ in eq. 4. We refer to the solution to the latter problem as $(\underline{ATE}(d), \overline{ATE}(d))$.*

**Implicit generative models.** To solve the partial identification problem, we use the expressive power of generative models to satisfy the distribution constraint in eq. 4. Choosing distance measures such as Jensen-Shannon divergence or Wasserstein metric results in models such as GANs or Wasserstein GANs (WGANs) [Goodfellow et al., 2014, Arjovsky et al., 2017]. The typical way to implement these models is to solve a minimax game between the generator and a discriminator. However, adding the ATD minimization/maximization term to the minimax loss function will result in unstable training. Instead, in our practical algorithm, we will use Sinkhorn Generative Networks (SGNs) that use Sinkhorn divergence $S_\epsilon$, a differentiable $\epsilon$-approximation of Wasserstein metric, as the distance measure between generated and true samples [Cuturi, 2013, Genevay et al., 2018, Feydy et al., 2019]. Due to the differentiability of Sinkhorn divergence, we will only need to train a generator network enabling us to sidestep much of the unstable minimax training in (W)GANs.

## 3  Related Work

This work builds upon partial identification and generative causal models.

**Partial identification.** Finding informative bounds on treatment effects has been well-studied in the existing literature ([Robins, 1989, Manski, 1990, Evans, 2012, Ramsahai, 2012, Richardson et al., 2014, Miles et al., 2015, Finkelstein et al., 2021, Zhang and Bareinboim, 2021a,b]). Balke and Pearl [1997] find the tightest possible bound for the discrete instrumental variable setting by converting it to a linear programming problem. For the backdoor setting and binary treatments, Makar et al. [2020] provide probabilistic upper/lower bounds on potential outcomes in the finite sample regime. Recently, Zhang et al. [2021] and Duarte et al. [2021] independently describe a polynomial programming approach to solve the partial identification for general causal graphs. They both use the notion of canonical SCMs to map the latent variables to the space of functions from treatment $T$ to outcome $Y$. Though they show their polynomial programming formulation finds the optimal bound, their approach is only applicable to discrete random variables with small support. In fact, the time complexity of their algorithm grows exponentially with the size of the support set of variables, making their algorithm intractable for continuous settings.

Gunsilius [2019] extends the commonly-used linear programming approach to partial identification of IV graphs with continuous treatments. They use a stochastic process representation of the variables and solve the linear programming via sampling. However, their method suffers from stability issues, as discussed in Kilbertus et al. [2020] and is only applicable for the IV setting. Kilbertus et al. [2020], Padh et al. [2022] parameterize the space of response functions by assuming them as linear combinations of a set of fixed basis functions. Then, they match the first two moments of observed distribution while minimizing/maximizing the ATE. However, they do not provide any theoretical guarantees on the tightness of their derived bounds.

---

[3]In the literature, partial identification is not concerned with sampling uncertainty and is defined population-wise for non-identifiable causal effects. However, in this paper, we abuse the terminology and use partial identification for both non-identifiable quantities and identifiable effects with finite samples.

Most similar to our work is Hu et al. [2021] who use generative adversarial networks (GANs) to match the observed distribution and search for response functions with maximum/minimum ATEs. They provide convergence guarantees for the instrumental variable causal graph with linear models. Their approach is also based on the minimax game between generator and discriminator, which can result in unstable training. Our work differs in a few important ways. We focus on partial identification of average derivatives and use that to find bounds over the ATE. Using this approach, we show that our derived bounds converge to the optimal bounds for linear SCM with general causal graphs, including both identifiable and non-identifiable settings. We use Sinkhorn divergence, a differentiable approximation of Wasserstein distance, to train our implicit generative models. Empirically, we find that this avoids the unstable training of GANs. Guo et al. [2022] studied the partial identification of ATE with noisy covariates. Their work is similar to our approach in that we both use a similar robust optimization formulation. However, they focus on identifiable causal graphs, where one can use adjustment formulas such as the Back-door formula and make parametric assumptions on the joint distribution of observed variables.

**Generative causal models.** [Goudet et al., 2017, Yoon et al., 2018, Kocaoglu et al., 2018, Sauer and Geiger, 2021] use generative models to capture a causal perspective on evaluating the effect of interventions on high-dimensional data such as images. They do not consider the problem of bounding treatment effects. Xia et al. [2021] introduced Neural Causal Models (NCMs) that leverages the universal approximability of neural networks to learn the SCM. Although it is not generally possible to learn the true SCM by training on the observational data, they prove that NCMs can be used to test the identifiability of causal effects and propose an algorithm to estimate identifiable causal effects. Their work's theory and empirical instantiation are in the context of discrete random variable datasets. Our work builds upon NCMs for partial identification with both continuous and discrete random variables.

# 4 Partial Identification using Implicit Generative Models

We explain our method to solve the partial identification problem in Def. 4 using implicit generative models. In subsection 4.1, we describe partial identification of ATDs as a constrained optimization problem using $\mathcal{G}$-constraint generative models [Xia et al., 2021]. Then, in subsection 4.2, we show that the solution to this constrained optimization problem converges to the optimal bounds on the ATD in infinite data samples. We prove our results for linear SCMs with general causal graphs, i.e., both identifiable and non-identifiable settings. Next, we propose our approach to extend the partial identification of ATDs to ATEs. Finally, we describe a practical algorithm to solve our distributionally-constrained optimization problem in subsection 4.3.

## 4.1 $\mathcal{G}$-constraint generative models

To solve the partial identification problem, we need to search over the set of all possible SCMs $\mathfrak{M}$. This is generally not feasible, as there is no constraint on the distribution of the latent variables $P_{\mathbf{U}}$, as well as the function family $\mathcal{F}$. Instead, we parameterize the space of all SCMs that are consistent with causal graph $\mathcal{G}$ using neural networks. More specifically, we use $\mathcal{G}$-constraint generative models:

**Definition 5** ($\mathcal{G}$-constraint Generative Models (Def. 7 in Xia et al. [2021])). *For a given causal graph $\mathcal{G}$, a $\mathcal{G}$-constraint generative model is a tuple $\mathcal{M}_{\mathcal{G}}^{\theta} = (\mathbf{V}, \hat{\mathbf{U}}, \mathcal{F}^{\theta}, \hat{P}_{\hat{\mathbf{U}}})$, where each $V_i \in \mathbf{V}$ is generated from*

$$V_i = f_{V_i}^{\theta}(\mathbf{pa}(V_i), \hat{\mathbf{U}}_{\mathbf{C}}) \text{ for } f_{V_i}^{\theta} \in \mathcal{F}^{\theta}, \tag{5}$$

*where $\mathbf{pa}(V_i)$ is the observed parents of node $V_i$ in $\mathcal{G}$ and $\hat{\mathbf{U}}_{\mathbf{C}} \in \mathbf{U}$ is the latent noise corresponding to maximal $C^2$-Component $\mathbf{C} \subseteq \mathbf{V}$ containing node $V_i$, i.e., each pair of variables in $\mathbf{C}$ have common latent parent nodes. In addition, $\hat{P}_{\hat{U}} \sim \mathtt{Unif}(0,1)$ for each $\hat{U} \in \hat{\mathbf{U}}$.*

$\mathcal{G}$-constraint generative models make the search over the set $\mathfrak{M}$ feasible by limiting it to generative models with *uniformly* distributed latent variables that are consistent with causal graph $\mathcal{G}$. In fact, in their Theorem 3, Xia et al. [2021] show that for *any* discrete SCM $\mathcal{M}^*$ with causal graph $\mathcal{G}$, there exists a $\mathcal{G}$-constrained generative model $\mathcal{M}_{\mathcal{G}}^{\theta}$ that generates the same causal effect, i.e., $Y_{\mathcal{M}^*(T=t)} = Y_{\mathcal{M}_{\mathcal{G}}^{\theta}(T=t)}$ a.s. Their proof technique, however, only works for SCMs with discrete variables. Here, we do not prove the expressiveness of $\mathcal{G}$-constrained generative models for continuous

SCMs. Instead, for simplicity and completeness of our theoretical results in subsection 4.2, we assume that the true SCM is a $\mathcal{G}$-constrained generative model itself. In our experiments, we empirically show that our results hold even for SCMs with different latent distributions, such as Gaussian noise.

**Assumption 1.** *The true SCM $\mathcal{M}$ is a $\mathcal{G}$-constrained generative model. In other words, there exist $\theta$ such that $\mathcal{M} = \mathcal{M}_{\mathcal{G}}^{\theta}$.*

Under Assumption 1, we reformulate the problem in eq. 4 using generative models, i.e.,

$$(\min_{\theta} \text{ATD}_{\mathcal{M}_{\mathcal{G}}^{\theta}}, \ \max_{\theta} \text{ATD}_{\mathcal{M}_{\mathcal{G}}^{\theta}}) \ \text{s.t.} \ P_{\mathcal{M}_{\mathcal{G}}^{\theta}} = P \tag{6}$$

In practice, we never have access to true distribution $P$ as we only observe a finite number of samples corresponding to the empirical distribution $P^n = \frac{1}{n} \sum_{i=1}^{n} \delta_{\mathbf{v}^{(i)}}$ for a given dataset $\{\mathbf{v}^{(1)}, \cdots, \mathbf{v}^{(n)}\}$. Also, the observed variables may be biased due to noisy measurements. Therefore, we reformulate the problem in eq. 6 as a constrained optimization problem. We choose the 1-Wasserstein metric as our distance measure, which naturally results in generative models such as WGANs. We will state our theory in subsection 4.2 based on this metric. However, in subsection 4.3, we will propose a practical algorithm that uses Sinkhorn divergence, a differentiable approximation of 1-Wasserstein distance, for more stable results. Our constrained optimization problem is as follows:

$$\left(\min_{\theta} \text{ATD}_{\mathcal{M}_{\mathcal{G}}^{\theta}}, \max_{\theta} \text{ATD}_{\mathcal{M}_{\mathcal{G}}^{\theta}}\right) \ \text{s.t.} \ W_1\left(P_{\mathcal{M}_{\mathcal{G}}^{\theta}}, P^n\right) \leq \alpha_n \tag{7}$$

where $\alpha_n$ is a hyper-parameter that specifies the level of tightness of the bounds. We denote the solution to eq. 7 as $(\widehat{\underline{\text{ATD}}}, \widehat{\overline{\text{ATD}}})$. In the case of noisy measurements, we need domain knowledge of how noisy the data is to determine the value of $\alpha_n$. Otherwise, we can use the finite-sample convergence rate of empirical Wasserstein distance to choose an appropriate value of $\alpha_n$ [Weed and Bach, 2019]. As our theoretical results are concerned with the infinite-sample case, we will assume that there exist values of $\alpha_n$ such that the true distribution lies within the Wasserstein ball.

**Assumption 2.** *For each $n \in \mathbb{N}$, there exist $\alpha_n > 0$ such that $W_1(P, P^n) \leq \alpha_n$.*

## 4.2 Theoretical guarantees and extension to ATEs

Assumptions 1 and 2 ensure that the bound derived by eq. 7 contains the true value of ATD. However, we do not know how informative/tight the derived bounds are. In fact, one can always return $(-\infty, +\infty)$ as one solution to partial identification. This part gives theoretical guarantees that our algorithm can result in tight bounds over ATD. In particular, we focus on linear SCMs and show that, under the infinite number of samples, our algorithm converges to the optimal bound $(\underline{\text{ATD}}, \overline{\text{ATD}})$ for both identifiable and non-identifiable causal graphs. See Appendix A for the proof.

**Definition 6** (Linear SCMs). *SCM $\mathcal{M} = (\mathbf{V}, \mathbf{U}, \mathcal{F}, P_{\mathbf{U}})$ is linear, if*

$$V_i = \mathbf{a}_{V_i}^{\top} \mathbf{pa}(V_i) + \mathbf{b}_{V_i}^{\top} \mathbf{U}_{V_i} \ \text{for vectors} \ \mathbf{a}_{V_i}, \mathbf{b}_{V_i} \in \mathcal{F} \tag{8}$$

**Theorem 1** (Tight Bounds). *Assume the dataset $\{\mathbf{v}^{(1)}, \cdots, \mathbf{v}^{(n)}\}$ is generated from a linear SCM. Then, under assumptions 1, and 2, the solution to the constrained optimization problem in eq. 7 converges to the optimal bound over the ATD in infinite samples, i.e., $\widehat{\underline{\text{ATD}}} \to \underline{\text{ATD}}$ and $\widehat{\overline{\text{ATD}}} \to \overline{\text{ATD}}$.*

Up until now, we have only focused on partial identification of ATDs. Here, we discuss how to extend our results to find bounds on ATEs. A naive solution is to replace ATD with ATE in eq. 7 and directly optimize it. However, as demonstrated in the experiments, this approach can result in non-informative bounds. In fact, ATE with continuous treatments is a non-regular quantity [Kennedy et al., 2017].

Instead, we claim that we can use the same generative model trained for partial identification of ATD to bound the value of ATE. In particular, we define a new objective function by uniform intervention on the treatment, which we call uniform average treatment derivative (UATD), and show that the solution to partial identification of UATD matches the solution to partial identification of ATEs.

**Definition 7** (UATD). *For an SCM $\mathcal{M}$, we define the uniform average treatment derivative (UATD) at interval $[t_0, d]$ as*

$$\text{UATD}_{\mathcal{M}}[t_0, d] := \mathbb{E}_{\mathbf{u} \sim P_{\mathbf{U}}} \left[ \mathbb{E}_{t \sim \textbf{Unif}[t_0, d]} \left[ \frac{\partial Y_t(\mathbf{u})}{\partial t} \right] \right] \tag{9}$$

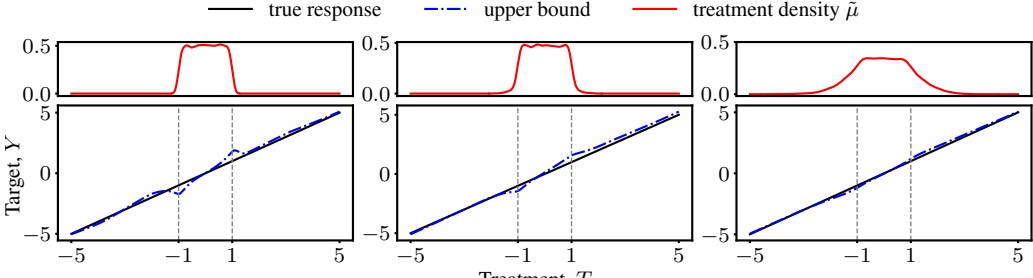

Figure 2: Maximizing the UATD between $T = -1$ and $T = 1$ with approximation densities $\tilde{\mu}$. The standard deviation of $\tilde{\mu}$ increases from left to right, resulting in smoother response curves. All three plots are trained on a linear data generating process $Y = T + \mathcal{N}(0, 1)$ using 5,000 samples

Now, we state our result on using average derivatives to solve partial identification of ATEs:

**Corollary 1.** *Let $\theta^*$ be the solution to the partial identification of UATD at interval $[t_0, d]$. Then, $\theta^*$ is also a solution to the partial identification of ATE$(d)$. If the true SCM $\mathcal{M}$ is linear, then the bound is tight, i.e., $\underline{\hat{ATE}}(d) \rightarrow \underline{ATE}(d)$ and $\overline{\hat{ATE}}(d) \rightarrow \overline{ATE}(d)$ as $n \rightarrow \infty$.*

The proof is given in Appendix B.

**Remark.** ATD and ATE are generally different quantities and finding bounds on one does not necessarily results in bounds on the other. In fact, ATE is not pathwise differentiable for continuous treatments [Díaz and van der Laan, 2013, Kennedy et al., 2017, Chernozhukov et al., 2018] (See Appendix C). We define UATD as a quantity to relate ATE and ATD. The formal definition of UATD in Def. 7 is the same as ATE up to a scale factor, and one would expect to see similar issues with ATE here as well. Therefore, we approximate UATD with a version that, instead of a uniform distribution of treatment within interval $[t_0, d]$ with zero density outside, the treatment distribution has continuous differentiable non-zero density defined over the *whole* support of $T$. More concretely,

$$\text{ATE}_{\mathcal{M}_{\mathcal{G}}^{\theta}}(d) \ \propto \ \text{UATD}_{\mathcal{M}_{\mathcal{G}}^{\theta}}[t_0, d] = \mathbb{E}_{\mathbf{u} \sim P_{\hat{\mathbf{U}}}} \left[ \int_{supp(T)} \frac{\partial Y_{\mathcal{M}_{\mathcal{G}}^{\theta}}(T = t)}{\partial t} d\mu(t) \right]$$

$$\approx \mathbb{E}_{\mathbf{u} \sim P_{\hat{\mathbf{U}}}} \left[ \int_{supp(T)} \frac{\partial Y_{\mathcal{M}_{\mathcal{G}}^{\theta}}(T = t)}{\partial t} d\tilde{\mu}(t) \right] \quad (10)$$

where $\mu$ is a the uniform measure within interval $[t_0, d]$ and $\tilde{\mu}$ is its approximation with a non-zero density over the full support of $T$. Choosing $\tilde{\mu}$ trades off between regularity of the response function and the approximation error. The more $\tilde{\mu}$ is close to the uniform measure $\mu$, the more irregularity we allow in the response curve and the less informative the bound will be, while the objective function is closer to ATE. Adding more density outside of interval $[t_0, d]$ imposes regularity on the response curve (Figure 2). It is important to note that we use the formal definition of UATD (with uniform treatment distribution) in Corollary 1 since, in linear SCMs, it is not possible for the generator model to attain arbitrarily large values in points $t_0$ and $d$ as the derivative of linear functions remains fixed outside of interval $[t_0, d]$. This is why we state our theoretical results based on the uniform intervention.

### 4.3 Our algorithm

We describe our algorithm to solve the optimization problem in eq. 7. We will focus on finding $\underline{\hat{ATD}}$, a similar approach can be taken for $\overline{\hat{ATD}}$. A general strategy is to convert the constrained problem to its unconstrained version using the method of Lagrange multiplier:

$$\min_{\theta} \max_{\lambda \geq 0} \text{ATD}_{\mathcal{M}_{\mathcal{G}}^{\theta}} + \lambda(W_1(P_{\mathcal{M}_{\mathcal{G}}^{\theta}}, P^n) - \alpha_n) \quad (11)$$

As the Wasserstein distance is not differentiable, we cannot directly use gradient descent to solve eq. 11. A common approach is to use the dual formulation of Wasserstein distance $W_1(P_{\mathcal{M}_{\mathcal{G}}^{\theta}}, P^n) = \max_{||q_\phi||_L \leq 1} \mathbb{E}_{P^n}[q_\phi(\mathbf{v})] - \mathbb{E}_{P_{\mathcal{M}_{\mathcal{G}}^{\theta}}}[q_\phi(\mathbf{v})]$ and solve eq. 11 using WGANs, a similar solution used

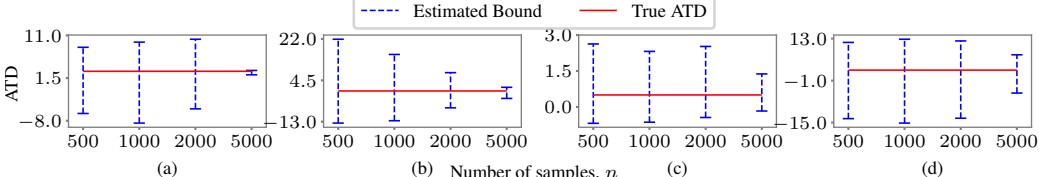

Figure 3: Our derived bounds on ATD for (a) linear Back-door, (b) Front-door, (c) linear IV and (d) leaky mediation settings. As the number of samples increases, our algorithm pin-points the ATD in identifiable settings and leads to tight bounds on it in non-identifiable cases.

in Hu et al. [2021]. However, this min-max-max formulation can result in unstable bounds as we show in our experiments. Instead, we use Sinkhorn divergence, a differentiable approximation to Wasserstein distance, as the measure of distance between distributions and solve the following:

$$\min_{\theta} \max_{\lambda \geq 0} \text{ATD}_{\mathcal{M}_{\mathcal{G}}^{\theta}} + \lambda(S_{\epsilon}(P_{\mathcal{M}_{\mathcal{G}}^{\theta}}, P^n) - \alpha_n) \tag{12}$$

To solve eq. 12, we need to evaluate $\text{ATD}_{\mathcal{M}_{\mathcal{G}}^{\theta}}$ and calculate its gradient w.r.t. $\theta$. As we are using $\mathcal{G}$-constrained generative models, we can calculate the value of $Y_{\mathcal{M}_{\mathcal{G}}^{\theta}(T=t)}(\mathbf{u})$ by hard intervention $T = t$, i.e., fixing the output of function $f_T^{\theta}$ as $t$ and computing $Y$ through a topological order of calculations. Then, we estimate $\text{ATD}_{\mathcal{M}_{\mathcal{G}}^{\theta}}$ as follows:

$$\text{ATD}_{\mathcal{M}_{\mathcal{G}}^{\theta}} \approx \frac{1}{n} \sum_{i=1}^{n} \frac{1}{\epsilon} \left[ Y_{\mathcal{M}_{\mathcal{G}}^{\theta}(T=t^{(i)}+\epsilon)}(\mathbf{u}^{(i)}) - Y_{\mathcal{M}_{\mathcal{G}}^{\theta}(T=t^{(i)})}(\mathbf{u}^{(i)}) \right] \tag{13}$$

where $\{t^{(i)}\}_{i=1}^{n}$ are samples from the treatment variable, and $\{\mathbf{u}^{(i)}\}_{i=1}^{n}$ are the latent variables generated from a uniform distribution. To choose an appropriate value of $\alpha_n$, we first train our generator without the ATD term until the Sinkhorn loss converges to some value and use that as our choice of $\alpha_n$. We then continue our training by adding the ATD term.

We note that using algorithms such as projected gradient descent to solve the constrained optimization problem requires us to project the weights of our network into the Wasserstein (Sinkhorn) ball in each step. This can be computationally infeasible, and current methods are mainly focused on special loss functions [Mohajerin Esfahani and Kuhn, 2018, Li et al., 2019, Wong et al., 2019]. Instead, we consider an alternating optimization procedure, in which we alternate between updating the gradients for the ATD and the Sinkhorn loss. The full details of our algorithm, its extension to ATEs, and the alternating optimization are described in Appendix D.

## 5 Experiments

We run our partial identification algorithms on a variety of simulated settings. We mainly focus on the synthetic data generating processes as the ground truth must be known to evaluate our derived bounds properly. Our primary goal is to show that using uniform average treatment derivatives instead of directly optimizing the average treatment effect will result in tighter and more stable bounds. We first run our algorithm to estimate bounds on the value of ATDs for both identifiable and non-identifiable causal graphs. We show that, as the number of samples increases, our algorithm converges to tight bounds over the true value of ATD (Figure 3). We then focus on partial identification of ATEs and demonstrate that using partial derivatives of the response function leads to more informative bounds, while being valid (Figures 4a, 4b).

Additionally, as a sanity check, we consider two binary-treatments datasets where the optimal bounds are known and show that our approach can reach the optimal solution. We also test our method on an ACIC dataset, a case study with real-world covariates, to illustrate the performance of our algorithm on higher-dimensional datasets. We include these additional experiments in Appendix E. Our implementation details, as well as hyper-parameters can be found in Appendix F.

### 5.1 Datasets and Baseline

**Datasets.** We consider various data generating processes for different causal graphs, including a linear SCM with three-dimensional covariates and a quadratic SCM with nonlinear interaction between

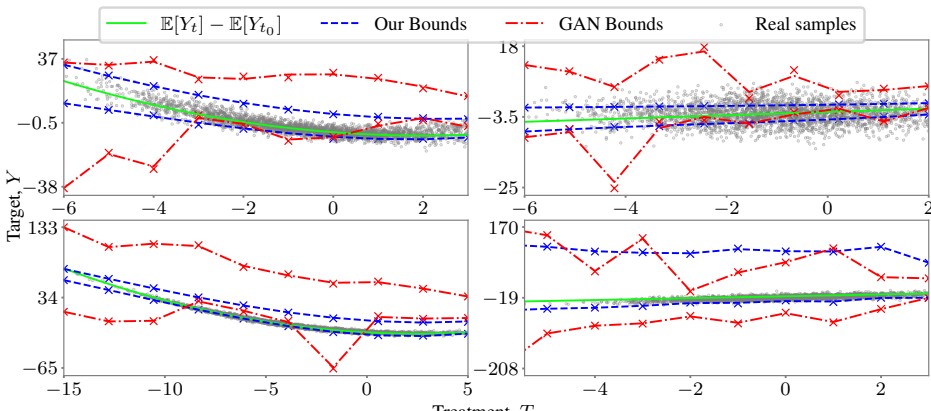

(a) GAN baseline. (top left) Nonlinear Back-door, (top right) linear IV, (bottom left) nonlinear IV, and (bottom right) leaky mediation settings. $t_0$ is chosen as the maximum treatment value in each data. Our derived bounds are tighter and more stable than the GAN baseline, which directly optimizes the ATE.

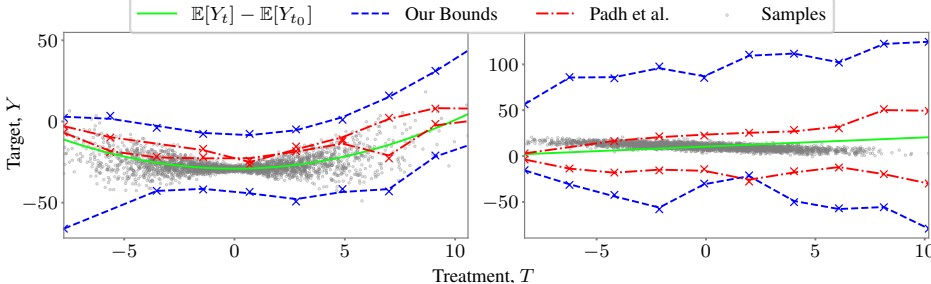

(b) Padh et al. [2022]. (left) Nonlinear IV, (right) linear IV with strong confounding. $t_0$ is chosen as the minimum treatment value in each data. While resulting in tighter bounds in the linear setting, the baseline leads to invalid bounds in the nonlinear dataset.

Figure 4: Comparing our results for partial identification of $\mathbb{E}[Y_t] - \mathbb{E}[Y_{t_0}]$ for 10 different values of treatment.

the covariates and treatment for the Back-door causal graph (Figure 1c), as well as a nonlinear SCM for the Front-door setting (Figure 1d). For the IV graph (Figure 1b), we use four different SCMs, two linear and two nonlinear datasets, based on the strength of the instrument variable and the confounding. Finally, we generate a two-dimensional linear dataset for the leaky mediation causal graph (Figure 1a). The full details of our data-generating processes can be found in Appendix G.

**Baselines.** Our first baseline is the algorithm in Hu et al. [2021] that directly optimizes the value of ATE using GANs. We use their default hyper-parameters with a tolerance of 0.0001. Similar to their experimental setup, we consider 50 intermediate solutions where the distance is within the tolerance and compute the bounds using the mean and one-sided confidence intervals.

We also compare our algorithm with the moment-matching method in Padh et al. [2022]. They parameterize the first two moments of the generated distribution and match them with the observed samples. They choose response functions as linear combinations of a fixed number of basis functions. We consider the neural basis functions for our experiments with their default hyper-parameters. In particular, we train a 3-hidden layer MLP with 64 neurons in each layer to learn the target variable given the treatment. We then use the activation of the $k$th neuron in the last hidden layer as the $k$th basis function. Instead of maximizing/minimizing $E[Y_{T=t}]$, we consider $E[Y_{T=t}] - E[Y_{T=t_0}]$ as our goal is to bound the ATE between two points. We leave other implementation details untouched.

### 5.2 Results

**Bounding average treatment derivatives.** We generate data with sample sizes $N = \{500, 1000, 2000, 5000\}$ from the nonlinear Front-door and linear Back-door SCMs (identifiable), as well as linear IV with strong confounding and leaky mediation settings (non-identifiable). We run our algorithm with ten different random seeds for each setting/sample size. Then, we choose

the five runs with the lowest tolerance parameter $\alpha_n$ and choose the upper (lower) bound as the maximum (minimum) value of the ATD within these five runs. Figure 3 shows our derived bounds. As expected, the algorithm is able to point-identify the value of ATD for identifiable scenarios as the number of samples increases (Figures 3 (a) and (b)). In non-identifiable cases, our algorithm leads to tight bounds containing the true value of ATD by increasing the number of samples as depicted in Figures 3 (c) and (d). This is in line with our results in Theorem 1.

**Bounding average treatment effects.** Here, we aim to demonstrate the effectiveness of using partial derivatives for bounding the ATE compared to the direct optimization approach. We consider six different settings and run our algorithm for 10 different values of treatment $\{t_i\}_{i=1}^{10}$ in each setting. We compute the value of ATE w.r.t. a fixed point $t_0$. For each value of $T$, we generate $N = 5{,}000$ samples and run each experiment five times. Then, we select the maximum (minimum) value of ATE within the five runs as the upper (lower) bound. We follow the same procedure for the baselines.

To find the bounds on ATE using our approach, we sample from a distribution with uniform density within $[t_0, t_i]$ and Gaussian tails outside of $[t_0, t_i]$, and maximize/minimize the partial derivatives. Figure 4a shows the effectiveness of this approach in comparison to the GAN baseline. Our algorithm produces stable and tight bounds containing the true value of ATE, while the GAN baseline, which relies on the direct optimization of ATEs, results in unstable loose bounds that may not include the true value of the treatment effect.

Figure 4b compares our algorithm with the method in Padh et al. [2022]. First, we find that their method do not contain the actual response curve in the nonlinear IV setting (left). This is not a surprising result; since they consider a fixed set of basis functions, the set of possible response curves is more restricted than ours. If the basis functions are not carefully designed to capture the true form of the response curve, the resulting bounds can be invalid, thus not containing the actual value of ATE even if there are regimes where the approach provides a tighter fit. On the other hand, our approach considers a larger family of response functions, enabling us to both capture shape and variation in the response curve while bounding the effects.

## 6   Conclusion, Limitations and Future Work

Our work introduces a novel method to estimate bounds on average treatment effects from observational data. Specifically, we propose optimizing the average treatment derivative, which in turn can be used to estimate the average treatment effect in treatment response curves. Empirically we find that the use of our method recovers known bounds on treatment effects in the discrete case and outperforms other methods based on implicit models for partial identification in the continuous case.

There remain several limitations of this work. Our work builds on the constrained optimization problem defined by Xia et al. [2021] instantiated in the context of the ATD. Developing new methods for function maximization/minimization approaches under distributional constraints remains an important direction for future work. Our work primarily uses carefully designed synthetic datasets to evaluate our method under different constraints on the data distribution. A larger-scale evaluation of our approach on real-world benchmarks will better help us assess the method's practicality. Moreover, the theory of our work is restricted to the linear SCM scenario. We have also made regularity assumptions throughout the paper, including Assumption 1, that the true SCM can be modeled using implicit generative models with uniform confounding distribution, as well as the approximation of UATD with regular treatment distributions. Consequently, practitioners must exercise caution when deploying this method when there are nonlinear or irregular structures among the random variables. Finally, we have stated our results (Theorem 1 and Corollary 1) without quantifying the finite-sample estimation uncertainty. One can use the existing theory of finite-sample convergence of empirical Wasserstein distance in Weed and Bach [2019] to extend our results with high-probability bounds.

## Acknowledgments

We thank Tom Ginsberg and Michael Cooper for many helpful discussions. We also thank David Alvarez Melis for his suggestion on using Sinkhorn divergence instead of Wasserstein distance. This research was supported by NSERC Discovery Award RGPIN-2022-04546 and a CIFAR AI Chair. Resources used in preparing this research were provided, in part, by the Province of Ontario, the Government of Canada through CIFAR, and companies sponsoring the Vector Institute.

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
