# OpenReview forum: "Partial Identification of Treatment Effects with Implicit Generative Models"
_NeurIPS.cc/2022/Conference — NeurIPS 2022 Accept_

### Official Review · Reviewer_sNpo · 2022-06-26

**Rating:** 8
**Confidence:** 4
**Soundness:** 4 excellent
**Presentation:** 4 excellent
**Contribution:** 3 good

**Summary:**

The manuscript proposes a novel method for bounding average treatment effects (ATEs) with continuous treatments by optimizing the average treatment derivative (ATD) using Sinkhorn generative networks. The authors empirically show this to result in stabler, more informative bounds than attempting to directly optimize the ATE itself.

**Questions:**

N/A

**Limitations:**

The limitations are well covered in §6.

**Strengths And Weaknesses:**

Partial effect identification with continuous treatments is an important and challenging problem, of great interest to methodologists and practitioners alike. The results here are quite impressive, including a theoretical guarantee in the case of linear SCMs and strong empirical performance under a range of different data generating processes. The manuscript is well-researched, building on a number of recent papers in this area. The writing is generally strong and easy to follow.

My only real critique is one readily acknowledged in the paper’s conclusion, namely that the empirical examples are drawn from a range of small-dimensional simulations rather than more challenging semi-synthetic or real-world datasets with more covariates. Generative modeling gets exponentially difficult in high dimensions, so methods based on GANs or other generative approaches may struggle in this setting. This is one motivation for Padh et al.’s (2022) partial identification approach, which relies instead on moment matching techniques. I am unsure whether their algorithm is publicly available for comparison, but if possible I think this would make a valuable baseline, especially on more complex data.

---

> ### Author Response · Authors · 2022-08-01
> **Response and clarifications to reviewer sNpo**
>
> Thank you for your encouraging and helpful comments. We are glad that you found our results impressive and the paper well-researched and easy to follow. We address your concerns below:
>
> > *My only real critique is one readily acknowledged in the paper’s conclusion, namely that the empirical examples are drawn from a range of small-dimensional simulations rather than more challenging semi-synthetic or real-world datasets with more covariates. Generative modeling gets exponentially difficult in high dimensions, so methods based on GANs or other generative approaches may struggle in this setting. This is one motivation for Padh et al.’s (2022) partial identification approach, which relies instead on moment matching techniques. I am unsure whether their algorithm is publicly available for comparison, but if possible I think this would make a valuable baseline, especially on more complex data.*
> >
>
> We agree that a limitation of the experimental setup is the low-dimensional simulations. However, part of the reason for these simulations is the impossibility of evaluation in real-world data, where we do not observe counterfactuals. As we described in the general response, we have included a new section (**Appendix E**) with additional experiments with a comparison to the method of Padh et al. We understand the reviewer's feedback on using more challenging semi-synthetic data (e.g., the covariates are high-dimensional, but the target variable is simulated). Although the existing literature on partial identification in the continuous setting (Hu et al., Padh et al., Kilbertus et al.) evaluate similar low-dimensional setups, we believe these methods' main benefit/limitations become clear in real-world high-dimensional datasets. We are currently working on such experiments, and if the paper gets accepted, we will include those in the camera-ready version. We appreciate it if the reviewer has any suggestion regarding those experiments. Also, note that our approach only needs to simulate the variables within the path between the treatment and outcome. For example, in the backdoor setting, our approach does not require generating the covariates, and it can use the available samples as the input of the target generation function. This is important since the treatment and outcome are often low-dimensional in practice.

---

> > ### Comment · Reviewer_sNpo · 2022-08-08
> > **Re: Response and clarifications**
> >
> > Many thanks to the authors for their thoughtful rebuttal and extended experiments. I was already positively inclined toward this submission, and will be revising my score upward to an 8 in light of author comments.

---

### Official Review · Reviewer_LsyV · 2022-07-12

**Rating:** 5
**Confidence:** 4
**Soundness:** 2 fair
**Presentation:** 3 good
**Contribution:** 2 fair

**Summary:**

The paper proposed to perform partial identification of average treatment effect using average treatment derivatives and implicit generative models. The paper argues that quantities like ATE can be hard to optimize due to certain regularities and resorting to average treatment derivatives and their expectations make things much more amenable to computation. Similarly, implicit generative models are also more amenable than GANs in modeling observed data distribution.

**Questions:**

Can you please clarify the points in the strengths and weaknesses section?

**Limitations:**

The paper discusses the limitations.

**Strengths And Weaknesses:**

Strengths:
+ The problem of partial identification with continuous variables is important.

+ The approach of using constrained optimization with implicit generative models is sound (under assumptions).

Weaknesses:

+ It is unclear why the idea of average treatment derivatives resolves the computational issues in partial identification. In particular, it seems like average treatment derivatives work in the proposed setting because it implicitly imposes regularity or smoothness assumptions like continuously differentiable log-density. Thus it is not the average treatment derivatives that are helping with the computational issues, but rather these additional regularity and smoothness assumptions that are helping with the computational issues. In principle, one should also be able to impose these assumptions to achieve computational gains without invoking average treatment derivatives.

+ Across the paper, there are (regularity) assumptions made, e.g.implicit generative models, in general, can only approximate a subset of all distributions; the paper also considers smooth densities for average treatment derivatives. While these assumptions make sense, they usually have non-trivial implications on the partial identification bounds. For example, it is well known that low-complexity unobserved confounders or functions (e.g. smooth functions) restrict how much the unobserved variables can induce confounding. Thus these regularity conditions are non-trivial, and their implications on the partial identification bounds need to be carefully and transparently studied. The validity of the resulting partial identification bounds must be qualified.

+ While the paper focuses on using implicit generative models for partial identification, it has limited applicability to usual sensitivity analysis assumptions like marginal sensitivity conditions. The algorithm can only work with the constraints that the implicit generative models induce a population distribution that is close to the observed data distribution. However, It appears very challenging to work with classical sensitivity assumptions: how could one impose the marginal sensitivity assumption (Tan, 2006) on the implicit generative model and obtain a partial identification bound? Without applicability to these standard sensitivity assumptions, the applicability of the proposed algorithm can be limited, since the ATE bound tends to be wide in practice.

+ Despite the asymptotic guarantees, the proposed algorithm cannot account for sampling uncertainty. It does not quantify the estimation uncertainty of the min and max ATE/ATD. (Also unlike L116, sampling variation does not motivate nor necessitate partial identification.)


Tan, Z. (2006). A distributional approach for causal inference using propensity scores. Journal of the American Statistical Association, 101(476), 1619-1637.

---

> ### Author Response · Authors · 2022-08-01
> **Response and clarifications to reviewer LsyV**
>
> We thank the reviewer for their detailed and thorough feedback. We are happy that you find our theoretical results sound. In the following, we address your questions.
>
> > *It is unclear why the idea of average treatment derivatives resolves the computational issues in partial identification. In particular, it seems like average treatment derivatives work in the proposed setting because it implicitly imposes regularity or smoothness assumptions like continuously differentiable log-density …. In principle, one should also be able to impose these assumptions to achieve computational gains without invoking average treatment derivatives.*
> >
>
> In the revised version, we have included a new section (**Appendix C)** to discuss the intuition behind using ATDs and their difference from ATEs. For a summary of that section, please see the general response. We agree with the reviewer that using UniformATDs implicitly imposes regularity assumptions that, in principle, are not exclusive to UniformATDs. Existing approaches in the literature of continuous treatment (e.g., Padh et al.) often enforce explicit assumptions on the functional family (e.g., linear combinations of fixed basis functions) or the distribution (e.g., only matching the first moments) to avoid irregularities. However, In our work, we do not make any parametric assumption on the functional families (except that they are parameterized using neural networks). Instead, we approximate the objective function ATE, which in principle is a pathological quantity, with the well-behaved objective function UniformATD as **one** way to impose regularities that allows us to trade-off between the smoothness of the response functions and the approximation error.
>
> > *Across the paper, there are (regularity) assumptions made, e.g. implicit generative models, in general, can only approximate a subset of all distributions; the paper also considers smooth densities for average treatment derivatives …. Thus these regularity conditions are non-trivial, and their implications on the partial identification bounds need to be carefully and transparently studied. The validity of the resulting partial identification bounds must be qualified.*
> >
>
> We agree with the reviewer that Assumption 1 (that the true SCM can be modeled using an implicit generative model) and the regularity assumption in the UATD objective can induce non-trivial implications on the bounds if the true SCM does not follow those. We have elaborated more on this limitation in the Conclusion section (L360) of the revised version. Analyzing the effect of violation of our assumptions on the validity of partial identification bound is an interesting direction, and we leave it as future work. We emphasize that, as Reviewer $\color{red}\text{3JcQ}$ has also pointed out, the trade-off between assumptions and the strength of the results is fundamental in causal inference. In practice, one needs domain knowledge to assess these assumptions, which are generally untestable only from data.
>
> > *While the paper focuses on using implicit generative models for partial identification, it has limited applicability to usual sensitivity analysis assumptions like marginal sensitivity conditions …. It appears very challenging to work with classical sensitivity assumptions: how could one impose the marginal sensitivity assumption (Tan, 2006) on the implicit generative model and obtain a partial identification bound? Without applicability to these standard sensitivity assumptions, the applicability of the proposed algorithm can be limited, since the ATE bound tends to be wide in practice.*
> >
>
> Assumptions on controlling the level of unobserved confounding in sensitivity analysis to get more informative bounds are indeed an important aspect of partial identification. In the linear case, this can be done by constraining the coefficients corresponding to latent variables. For the non-linear case, defining those assumptions as differentiable quantities and using them as a component of generative models is an interesting direction we leave for future work. Note that we see our work as an independent approach with its own set of assumptions to tackle this problem.
>
> > *Despite the asymptotic guarantees, the proposed algorithm cannot account for sampling uncertainty … (Also unlike L116, sampling variation does not motivate nor necessitate partial identification.)*
> >
>
> While the current presentation of the paper ignores sampling uncertainty, we argue our algorithm **is capable** of incorporating the uncertainty by providing high probability bounds on ATE. We refer the reviewer to the general response for a detailed discussion. We have also added a footnote to L116 in the revised version to clarify that the partial identification literature is not concerned with sampling variation but we abuse the terminology and refer to both sampling uncertainty and non-identifiability as partial identification.

---

> > ### Comment · Reviewer_LsyV · 2022-08-07
> > **Response**
> >
> > Thank you for your response and explanation. I have read the rebuttal.
> >
> > I understand the point about the tradeoff between the strengths of assumptions and the complexity of the computation. However, my point is more about positing explicit assumptions as opposed to implicit assumptions. Since partial identification intervals can be sensitive to even regularity assumptions, these assumptions need to be imposed explicitly --- it is a situation different from classical estimation where regularity conditions play a smaller role in the resulting estimate than in partial identification. In this sense, avoiding explicit parametric assumptions is actually a downside and undesirable if it is just replacing them with implicit regularity conditions (as this paper does). We'd always favor transparent assumptions in comparing the proposed method in the paper and existing partial identification work like Padh et al.
> >
> > After all, my evaluation remains the same.

---

### Official Review · Reviewer_jbMK · 2022-07-12

**Rating:** 6
**Confidence:** 2
**Soundness:** 3 good
**Presentation:** 2 fair
**Contribution:** 3 good

**Summary:**

This work proposed a new method for partial identification. The new method uses GAN for modeling, with the novelty of replacing the usual objective function by a new one that is based on average treatment derivatives, which is shown in this paper to have superior performance than the vanilla average treatment effects. The core is by putting assumption of G-constraint generative models with uniform distribution of latent variables, which leads to Equation (9) connecting ATE with ATD.

**Questions:**

1. In experiments the author seems to compare only with one baseline -- are there any other baselines that can be used? For this paper, experimental evaluations are very important, but the current experiment section seems a little bit insufficient.

2. It is not clear to me why using ATD is better than ATE? Some intuition should be added in the paper.

**Limitations:**

Yes

**Strengths And Weaknesses:**

Originality:
I am not very familiar with related literature. Based on section 3, I think the paper has sound originality compared with existing works.

Quality:
The quality of this work is acceptable.

Clarify:
The presentation of this paper is good.

Significance:
The new objective function (ATD) proposed in this work is not dramatically different from the existing ones, but seems to result in great improvement in experiments. One concern on the experiment section is raised in the next section.

I believe the main strength of this paper is the author's clever use of the existing ideas in literature, especially the concept of G-constraint generative models in Xia et al [2021]. This observation itself leads to a simple but seems effective method. This contribution is significant, beyond that other technical ones are less so.

Some minor comments:
1. In Line 241, the convergence of a interval is not defined
2. Assumption2 seems a little bit weird — is it a trivial assumption?

---

> ### Author Response · Authors · 2022-08-01
> **Response and clarifications to reviewer jbMK**
>
> Thank you for your thoughtful feedback. We are encouraged that you find this paper original and the idea of using G-constrained generative models for partial identification a significant contribution. We address your questions below.
>
> > *In Line 241, the convergence of a interval is not defined.*
> >
>
> In the revised version, we have fixed this issue in both Theorem 1 and Corollary 1 by writing the convergence of an interval in terms of the convergence of the upper and lower bounds.
>
> > *Assumption 2 seems a little bit weird — is it a trivial assumption?*
> >
>
> Assumption 2 is not trivial since, in practice, we can only have high-probability bounds on the value of $W_1(P, P^n)$. We have assumed that the true distribution always lies within the Wasserstein ball for some value of $\alpha_n$ solely to avoid technicalities concerned with finite samples. However, our results will still hold without Assumption 2 in a probabilistic sense. We refer the reviewer to the general response for a more detailed discussion.
>
> > *[Q1] In experiments the author seems to compare only with one baseline -- are there any other baselines that can be used? For this paper, experimental evaluations are very important, but the current experiment section seems a little bit insufficient.*
> >
>
> Thank you for pointing this out. As far as we know, two baselines in the literature work with general causal graphs (Hu et al. and Padh et al.). We asked Padh et al. for their implementation and included additional experiments in **Appendix E** of the revised version. Please also see the related section in the general response.
>
> > *[Q2] It is not clear to me why using ATD is better than ATE? Some intuition should be added in the paper.*
> >
>
> We have provided a new section in the supplementary **(Appendix C)** of the revised version for more discussion on the intuition behind using ATDs instead of ATEs. We have also included a summary of that section in the general response.

---

> > ### Comment · Reviewer_jbMK · 2022-08-08
> > **Author Response Acknowledgement**
> >
> > Thank you to the authors for the updated manuscript and detailed response.
> >
> > From mathematics point of view, replacing ATE by UATD is straightforward, but it is interesting to see that it has resulted in much better performance for generative models. This is of course coupled with the implicit smoothness assumptions put by the derivatives in UATD, but the property of generative models also plays a role here. I think this is an important point and should be discussed explicitly, and in more detail in the main paper, instead of in the appendix only.
> >
> > Otherwise I am satisfied with the authors' response and am happy to improve my evaluation to 6.

---

> > > ### Author Response · Authors · 2022-08-09
> > > **Response to reviewer jbMK**
> > >
> > > We thank the reviewer for their encouraging comment. We are happy that you are satisfied with our response. Regarding the smoothness assumptions in UATD and the difference with ATE, we agree that more detail should be added to the main manuscript. If the paper gets accepted, we will move Appendix C to the main paper with additional experiments as an extra page is allowed in the camera-ready version. Also, if you are willing to increase the evaluation, could you please edit the initial rating? Thanks again for your comment, and please let us know if you have more questions.

---

> > > > ### Comment · Reviewer_jbMK · 2022-08-09
> > > > **Changed evaluation**
> > > >
> > > > Thanks. The evaluation has changed in my initial review.

---

### Official Review · Reviewer_3JcQ · 2022-07-13

**Rating:** 6
**Confidence:** 4
**Soundness:** 3 good
**Presentation:** 3 good
**Contribution:** 2 fair

**Summary:**

This paper proposes a new method the partial identification of causal effect for when the variables might be continuous. There has been a lot of progress in the identification of causal effect in the discrete and finite settings, but bounding for continuous treatments is relatively new.

Similar to most recent work on bounding treatment effects for continuous variables, the paper defines a generative model and bounds effect by minimizing and maximizing the quantity of interest (in this case Average Treatment derivatives (ATD)) with the constraint that the the generated data should match the observed data. The authors then show how bounding ATD also helps to bound ATE without doing a separate optimization, under certain assumptions such as the distribution of the confounding being uniform for example. The authors then compare their work to to the work of Hu. et al and show that they get tighter bounds.

**Questions:**

* Is it a new idea to use ATD to bound ATE or has this been done before?
* l544 (appendix). 'Our proof can be extended to multi-dimensional random variables by assuming each dimension as a separate variable.'. Could the authors elaborate on this a bit?
* Figure 2: Have the authors also done this experiment with the Hu et al. comparison?
* Is there a particular reason why Padh et al. was not used as one of the baselines as well?


**Limitations:**

The authors have generally addressed the limitations well. Though I would suggest to add Assumption 1 as a limitation as I have detailed in the main section of the review.

**Strengths And Weaknesses:**

**Strengths**
I find the paper to be generally well written and easy to follow. The assumptions are also clearly stated. And while the use of a distributionally constrained optimization formulation for partial identification of continuous treatment effects is not new (Hu et al., Padh et al., Kilbertus et al.), I find the idea to use ATD to bound ATE quite nice and certainly relevant and interesting to the community. The authors also provide a very nice overview of the related work. It is also worthwhile and interesting that the authors provide a proof of validity and tightness of the bounds for continuous variables, even if in limited settings and under strong assumptions.

**Weaknesses**
* Assuming a uniform distribution for the confounders, and Assumption 1: In justifying assumption 1, the authors mention that a similar result holds for discrete variable as shown by Xia et al. and the authors believe  'a similar approach can be used to show their results for general SCMs'. However, this is far from trivial. Proofs in causal inference which assume discrete variables generally revolve around some kind of counting arguments, and are not easily extendible to the continuous setting, and might not necessarily hold. It is not obvious that $\hat{P}_{\hat{U}} \sim Unif(0, 1)$ is reasonable to assume when the variables are not just continuous but also high dimensional. However, granting that a lot of causal inference is a tradeoff between assumptions and the strength of statements we can make, perhaps this can be considered a relatively small weakness. However, it would be useful to mention this clearly in the limitations section at the very least.

Some comments
* l36: 'Padh et al. make parametric assumptions on the distribution of treatment-response functions'. From my understanding, this might be a slight mischaracterization since they don't seem to make assumptions on the distribution of the response functions but rather parametrize the mean and covariance of the response functions.
* l173: Section 'Generative causal models' and comparison to NCM: The authors mention that their work 'can be seen as an extension of NCM'. However Xia et al. provide a sound and complete algorithm for effect identifiability in NCMs, which is why the NCMs are interesting. This paper provides a very limited setting in which their bounds are valid, and while making assumptions (such as Assumption 1) which would be quite non-trivial to justify (see weakness section). Therefore, in my opinion it might be better not to call this method an extension of NCMs.

Overall, I find that this work contributes some new new ideas to the topic of partial identification which could be interesting to the community.

---

> ### Author Response · Authors · 2022-08-01
> **Response and clarifications to reviewer 3JcQ**
>
> Thank you so much for your detailed and helpful comments. We are encouraged that you appreciate the contribution of the work and find the paper well written and interesting. In the following, we address your questions.
>
> > [Weaknesses] *Assuming a uniform distribution for the confounders, and Assumption 1*
> >
>
> We agree with the reviewer that most of the proofs in (finite) discrete causal inference are mainly based on counting arguments, e.g., one of the main techniques is to use canonical SCMs, which partition the (possibly continuous) latent space into a “finite” number of subsets. In the revised version, we have removed the part on "a similar approach can be used to show their results for general SCMs" in L206. We have expanded the limitation (L360) to  better emphasize the limitations of Assumption 1 and the need for a careful assessment of this assumption in practical applications with continuous treatment.
>
> > [Comment] *'Padh et al. make parametric assumptions on the distribution of treatment-response functions'. From my understanding, this might be a slight mischaracterization since they don't seem to make assumptions on the distribution of the response functions but rather parametrize the mean and covariance of the response functions.*
> >
>
> Thank you for pointing this out. This is indeed an error in line 36 regarding the parametric assumptions on the distribution of treatment-response in Padh et al. We change that line to ‘parametric assumptions on the **family** of treatment-response functions’ since they assume the family of feasible functions as linear combinations of some fixed set of basis functions. Padh et al. only parameterize the mean and covariance of the distribution over response functions since their experiments work with a notion of distance that is only based on the first two moments. However, for a general distribution distance, as they explicitly state, they need to sample from a fully-specified version of the distributions to match the observed samples with their generated samples.
>
> > [Comment] *Section 'Generative causal models' and comparison to NCM: The authors mention that their work 'can be seen as an extension of NCM'. However Xia et al. provide a sound and complete algorithm for effect identifiability in NCMs, which is why the NCMs are interesting … Therefore, in my opinion it might be better not to call this method an extension of NCMs.*
> >
>
> This is a fair point. We intended to emphasize that we are using NCMs, i.e., G-constrained generative models as the building blocks of our work. Therefore, we have changed this sentence to ‘Our work builds upon NCMs for …’ to avoid mis-conceptions.
>
> > *[Q1] Is it a new idea to use ATD to bound ATE or has this been done before?*
> >
>
> Yes. Note that ATD (aka average partial effect) is not a new quantity [Powell et al., 1989, Wooldridge, 2005, Rothenhäusler and Yu, 2019]. However, as far as we know, the idea of using ATD to estimate/bound ATE has not been explored before.
>
> > *[Q2] l544 (appendix). 'Our proof can be extended to multi-dimensional random variables by assuming each dimension as a separate variable.'. Could the authors elaborate on this a bit?*
> >
>
> We explain the multi-dimensional case by a simple example. Assume the causal graph is given by $T \to Y$ and both $T$ and $Y$ are high dimensional, i.e., $T = (T_1, \cdots, T_m)$ and $Y = (Y_1, \cdots, Y_k)$. Then, assuming each dimension as a separate variable results in the new causal graph with edges $T_1 \to Y_1, T_1 \to Y_2, \cdots, T_1 \to Y_k, T_2 \to Y_1, \cdots, T_m \to Y_k$. Note that this new graph is equivalent to the original one and we only consider that for the sake of proof. In our implementation, we model each variable as a vector.
>
> > *[Q3] Figure 2: Have the authors also done this experiment with the Hu et al. comparison?*
> >
>
> No, because their objective function is different. The GAN method from Hu et al. optimizes the direct objective **ATE**, while Figure 2 shows the convergence to the true bound on the value of **ATD**. Though it is possible to use their method to estimate ATD by changing the objective function. Furthermore, we note that an important distinction between our algorithm and theirs is the objective function itself: we use Uniform ATDs (a form of ATD in which we uniformly intervene on values of treatment within an interval) to estimate ATE, while they directly optimize average treatment effect.
>
> > *[Q4] Is there a particular reason why Padh et al. was not used as one of the baselines as well?*
> >
>
> We did not include Padh et al. as a baseline since the code was not publicly available. We asked the authors for their implementation and have included a new section (**Appendix E**) as additional experiments. Please also see the general response for a detailed discussion.

---

> > ### Comment · Reviewer_3JcQ · 2022-08-09
> > **Response**
> >
> > Thank you for the answers to the questions. I am encouraged by the minor changes in the text made by the authors and I believe that they address well my comments on comparisons to SCMs and assuming scalar and uniform gaussian confounders.
> >
> > Regarding high-dimensional treatments, I'm not sure that a complete bi-partite graph between two high dimensional variables $T$ and $Y$ considering each dimension as a separate node as constructed in your answer is equivalent to the original graph. Such a construction could imply dependences not implied in the original graph. For instance, say $T = (T_1, T_2, T_3)$ and $M = (M_1, M_2, M_3)$, in your example. Then the following distinct set of edges all lead to the graph $T \rightarrow Y$.
> > $E_1$: $\\{ T_1 \rightarrow M_1, T_2  \rightarrow M_2, T_3  \rightarrow M_3 \\}$
> > $E_2$: $ E_1 \cup \\{T_1 \rightarrow T_2, M_1 \rightarrow M_2 \\}$
> > $E_3$: Construction as defined by the authors
> > .. and it goes on of course. All of these graphs are different and imply different relationships at a node level below the high-dimensional level. However, as far as I can see they all imply the same high-dimensional relationship between $T$ and $Y$, which is the only information we really have from the graph. Of course, the same argument also applies to graphs with more than 2 nodes, where the problems only get worse. I would be happy to read the authors' take on this.

---

> > > ### Author Response · Authors · 2022-08-09
> > > **Thank you**
> > >
> > > Thank you for your reply. We are glad that our response addressed most of your questions.
> > >
> > > Regarding the high-dimensional variables, we admit there was a mistake in our response regarding the "equivalence" of the graphs. Indeed, your reasoning is correct that there might be multiple graphs with possibly different node-level dependence structures that have the same structure at the high-dimensional level. We appreciate your detailed evaluation. Here, we refine our original answer:
> > >
> > > Suppose a general causal graph with possibly high-dimensional variables. There exist a finite number of graphs with one-dimensional nodes with the same dependence structure in the original graph. To construct these graphs, it is sufficient to have at least one edge from the parent node cluster to the child node cluster (other relationships can be arbitrary). We assume one of these one-dimensional causal graphs represents the "true" independence structure between variables. Since our proof works for every causal graph with the one-dimensional case, it will also work for the "true" one. We emphasize that we did not include the high-dimensional proof to avoid extra technicalities that can harm the readability of the proof.
> > >
> > > We hope our comment addresses your question. Please let us know if we can answer any further questions that might improve your assessment of the work.

---

> > > > ### Comment · Reviewer_3JcQ · 2022-08-09
> > > > **High-dim variables**
> > > >
> > > > Thanks for your response. Sure, I see now that the proof would work for the 'true' unrolled one-dimensional causal graph. However, I have some semi-serious concerns about this and do not quite see how it is simply a 'matter of technicality' as you say
> > > > * The proof for the high-dimensional case then assumes that you know the true dependence between each node. While I understand and agree with the assumption that the general causal graph between high-dimensional variables is given, it is a much stronger assumption that the node-level graph is given, and a lost less justifiable as well, as I'm sure you will agree. Unless I'm missing something, to say that the proof works for high-dimensional variables, you would have to add this assumption as Assumption 3. I'd be very happy to be corrected about this since I do like the paper as my review suggests.
> > > > * A similar argument then applies to the experiments as well. I assume that you run the experiments using the high-level graph and not the low-level graph, in which case they would not provably converge if I understand correctly?
> > > >
> > > > Assuming all of the above I say is correct, this does challenge a bit the theoretical claims of the paper. At the end of the day, even assuming that the theoretical results are only valid for the scalar case, I still believe that the paper is interesting. Partial identification in the continuous case is hard. Saying that the proof works for the scalar case, and the method seems to work empirically for the multi-dimensional case is not unreasonable in my opinion. However, it seems unhelpful to simply say that the high-dimensional proof is not shown for technical reasons if there are the above raised issues to consider (though if I am shown to be wrong about the above points then it is fine of course).
> > > >
> > > > P.S. My apologies for bringing this up this late. This part was not clear to me in the original submission (hence my question) or I would have raised this point sooner.

---

> > > > > ### Author Response · Authors · 2022-08-09
> > > > > **Response to the High-dim variables case**
> > > > >
> > > > > Thanks for the feedback and the engagement! The crux of this question is whether one needs to know the fully unrolled causal graph.
> > > > >
> > > > > We emphasize that our definition of the "optimal" bound depends on the "given" causal graph $\mathcal{G}$ (which we assume is all the available knowledge from a domain expert).
> > > > > In other words, if $\mathcal{G}$ is not informative about the relationship between each dimension, we expect the optimal bound to be less informative.
> > > > >
> > > > > * In theorem 1, we show that our algorithm converges to the optimal bound, i.e., the tightest bound *given* the information of the causal graph.
> > > > > * In our experiments, the data generating process of the multi-dimensional cases always depends on all the dimensions, i.e., it is a complete bipartite graph. Therefore, the bounds are the tightest ones.
> > > > >
> > > > >
> > > > > To see why, let us define each dimension of a $m$-dimensional variable $V\_i$ as nodes $V\_{i1}, V\_{i2}, \cdots, V\_{im}$.
> > > > >
> > > > > We consider two cases:
> > > > >
> > > > > **Case 1**: No inter-dependence between nodes of a multi-dimensional variable.
> > > > >
> > > > > In this case, for each $V\_i$, there is no causal edge between $V\_{ij}$ and $V\_{ik}$. Therefore, *eq. 13* still holds, i.e.,
> > > > > $$V\_{ij} = {\bf \theta}\_{V\_{ij}}^\top \bf{pa}(V\_{i}) + {\hat{\bf \theta}}\_{V\_{ij}}^\top \hat{\bf{U}}\_{C\_{i}}$$
> > > > >
> > > > > where ${\bf \theta}\_{V\_{ij}} \in \mathbb{R}^{|\bf{pa}(V\_i)|\cdot m}$ and $\bf{pa}(V\_{i}) \in \mathbb{R}^{|\bf{pa}(V\_i)|\cdot m}$ is the column vector after concatenating all the $m$-dimensional parents together.
> > > > >
> > > > > Note that here we do not know the true causal graph (i.e., which dimension of each parent is the *actual* parent of node $V\_{ij}$). However, since we allow the elements of ${\bf \theta}\_{V\_{ij}}$ to be in $\mathbb{R}$, they can take value $0$.
> > > > > Accordingly, in *Lemma 2*, $\bf{V}\_{\theta}$ will be a $d\times m$ matrix, and ${\bf A}(\theta)$ a  $d\times m \times |dim(\hat{\bf{U}})|$ tensor.
> > > > >
> > > > > In other words, in this case, the assumption that $V\_i$ is a general (linear) function of its parents will include the true causal graph.
> > > > >
> > > > >
> > > > > **Case 2**: There are possible causal edges between $V\_{ij}$ and $V\_{ik}$. We can still convert this case to case 1 without loss of generality. For instance, assume both $T$ and $Y$ are two-dimensional, and the causal graph is as follows:
> > > > > $$T\_1 \to Y\_1, T\_2 \to Y\_2, Y\_1 \to Y\_2$$
> > > > >
> > > > > And the SCM is as follows:
> > > > > $$Y\_1 = \theta\_1 T\_1 + \text{noise}$$
> > > > > $$Y\_2 = \theta\_2 T\_2 + \theta\_3 Y\_1 + \text{noise}$$
> > > > >
> > > > > Then, we can convert the equation of $Y\_2$ to the following:
> > > > > $$Y\_2 = \theta\_2 T\_2 + \theta\_3 \cdot \theta\_1 T\_1 + \text{noise}$$
> > > > >
> > > > > Note that since we only intervene on $T$, the interventional expectation (the quantity we bound) of $Y\_2$ will be the same for both SCMs.
> > > > >
> > > > > (We assume that either the intervention happens on all of $T$ instantaneously or there is no causal edge between dimensions of the treatment variable in the multi-dimensional treatment case. Otherwise, one must re-define the treatment variable into multiple treatment variables).
> > > > >
> > > > > In summary, to directly address your bullet points:
> > > > > Associated with each graph is a notion of the tightest bound for that graph. The proof shows convergence for any "given" graph i.e.
> > > > > our definition of the "optimal" bound depends on the given causal graph (which is the right quantity to look at since it is all the available information a practitioner has).
> > > > > Since the node-level graph has strictly more conditional independence statements than the aggregated graph, we expect that knowing the former will result in tighter bounds on the interventional expectations.
> > > > >
> > > > > One can ask about the relationship between our algorithm's result in the multi-dimensional case with the *aggregate* graph and its relationship to the optimal bounds associated with the true node-level graph. What we know is that the search space over the *aggregate* graph
> > > > > will contain the solution to the optimal bounds with the true node-level graph (i.e., the optimal bound over the aggregate graph includes the optimal bound over the true node-level graph).
> > > > >
> > > > > That said, your question does raise a valid point that the extension to the multi-dimensional case is more than a technical note that requires careful thought along these cases (with at least the assumption above).
> > > > > We will rephrase the sentence in the manuscript to point out some of the subtleties needed for extending to the multi-dimensional case.

---

### Author Response · Authors · 2022-08-01
**General Response**

We thank all the reviewers for their helpful and thorough feedback. We are happy that they found our work interesting and relevant (Reviewer $\color{red}\text{3JcQ}$), original and a significant contribution ($\color{blue} \text{jbMK}$), well-written and easy to follow ($\color{red}\text{3JcQ}$, $\color{green}\text{sNpo}$), sound ($\color{orange}\text{LsyV}$), and our theoretical and empirical results impressive and strong ($\color{green}\text{sNpo}$). In the following, we first address the common concerns on accounting sampling uncertainty, the intuition behind our method, and the experimental setting. We will answer each reviewer's specific questions and concerns separately.

**Sampling uncertainty and Assumption 2 ($\color{blue} \text{jbMK}$, $\color{orange}\text{LsyV}$)**

Throughout the paper, we have stated our results (Theorem 1 and Corollary 1) without quantifying the estimation uncertainty of the bounds on Average Treatment Effect (ATE) and Average Treatment Derivative (ATD). More concretely, we have assumed that for each number of samples $n \in \mathbb{N}$, there exists some value of $\alpha_n > 0$ that the Wasserstein distance of the true distribution $P$ with the empirical one $P^n$ is smaller than $\alpha_n$ and $\alpha_n \to 0$ as $n \to \infty$. This assumption is generally not valid since we can only define probabilistic bounds on the Wasserstein distance in finite samples. We made this assumption solely to avoid technicalities that can harm the readability of the paper and its primary purpose.

That said, we argue that similar results hold **without** Assumption 2. According to Proposition 10 in Weed and Bach [2019], we have

$\mathbb{E}\left[W_1(P, P^n)\right] \leq C_1 n^{-1/d_n}$

for some value of $C_1$ and $d_n$ that only depend on $n$. Then, from Proposition 20 [Weed and Bach, 2019]:

$\mathbb{P} \left[W_1(P, P^n) < \mathbb{E}\left[W_1(P, P^n)\right] + t \right] > 1 - exp(-2nt^2)$

Combining the above inequalities:

$\mathbb{P} \left[W_1(P, P^n) < C_1 n^{-1/d_n} + t \right] > 1 - exp(-2nt^2)$

Defining $\delta = exp(-2nt^2)$, we will have

$\mathbb{P} \left[W_1(P, P^n) < \alpha_n(\delta) \right] > 1 - \delta$

where $\alpha_n(\delta) = C_1 n^{-1/d_n} + \sqrt{\frac{\log 1/\delta}{2n}}$. Therefore, instead of Assumption 2, we have a high probability bound on the value of Wasserstein distance for some specific choice of $\alpha_n$. We can then use this bound to derive confidence intervals on the partial identification bounds in the finite sample setting. We will expand on this avenue in future work.

**Intuition behind ATDs ($\color{blue} \text{jbMK}$, $\color{orange}\text{LsyV}$)**

We have included a section in the supplementary **(Appendix C)** of the revised version to explain the difference between ATE and ATDs. Here, we provide a summary of that section. Suppose we have a simple identifiable causal graph (e.g., $T \to Y$) and we are interested in bounding $\mathbb{E}\left[Y_{T=d}\right] - \mathbb{E}\left[Y_{T=t_0}\right]$. Since we have only access to finite data points, the probability of observing data points exactly at $T=d$, $T=t_0$ is zero. Now, using an expressive generative model like GANs, we are able to match the finite dataset exactly (in terms of any distribution distance including Wasserstein distance), while creating arbitrarily large values at $T=d$ and $T=t_0$. See **Figure 4a** (in Appendix C of the revised version) for a demonstration. This means that ATE, in the continuous treatment case, is a pathological quantity and finding informative nonparametric bounds on it is impossible. In our approach, we use UniformATD (UATD) instead of ATEs. We formally define UATDs (Definition 7) as the average partial derivatives of the response function w.r.t. a uniform treatment distribution in interval $[t_0, d]$. This definition equals ATE up to a scale factor and, therefore, it will have similar pathological issues as ATE. Instead, we approximate UATD (ATE) with a version that, instead of uniform distribution in $[t_0, d]$ with zero density outside, the treatment distribution has continuous differentiable non-zero density defined over the **whole** support of $T$. More concretely, we have

$\text{ATE}\_{\mathcal{M}\_\mathcal{G}^\theta}(d) \propto \text{UATD}\_{\mathcal{M}\_\mathcal{G}^\theta}[t\_0, d] = \mathbb{E}\_{u \sim P\_{\hat{{U}}}}\left[\int\_{supp(T)} \frac{\partial Y\_{\mathcal{M}\_\mathcal{G}^\theta(T=t)}}{\partial t} d\mu(t) \right] \approx \mathbb{E}\_{u \sim P\_{\hat{{U}}}}\left[\int\_{supp(T)} \frac{\partial Y\_{\mathcal{M}\_\mathcal{G}^\theta(T=t)}}{\partial t} d\tilde{\mu}(t) \right]$

where $\mu$ is the uniform measure within $[t\_0, d]$ and $\tilde{\mu}$ is its approximation with a non-zero density over the full support of $T$. Choosing density $\tilde{\mu}$ allows us to trade off between regularity of the response curve and the approximation error. See **Figure 4b** in Appendix C for more intuition.

---

> ### Author Response · Authors · 2022-08-01
> **General Response (Cont.)**
>
> **Experiments and baselines  ($\color{green}\text{sNpo}$, $\color{red}\text{3JcQ}$ , $\color{blue} \text{jbMK}$)**
>
> Regarding the baselines, we are only aware of two existing methods that work with general causal graphs, the GAN method by Hu et al. and the moment matching method by Padh et al. We compared our algorithm to the GAN method as we have similar choices of the family of response functions (i.e., general neural networks). Also, there was no publicly available implementation for the Padh et al. baseline. Nevertheless, we asked the authors to share their implementation. We included additional experiments in **Appendix E** of the revised version to compare with **Padh et al**. Since they assume the response curves as linear combinations of a fixed set of basis functions, their search space is strictly smaller than ours naturally resulting in tighter bounds. However, this can lead to **invalid bounds** that do not include the actual value of ATE when the basis functions do not capture the true form of the response function. Please see **Figure 5 (left)** in Appendix E for more details.

---

### Comment · Area_Chair_9kMc · 2022-08-08
**Discussion with Authors**

Dear Reviewers! Thank you so much for your time on this paper so far.

The authors have written a detailed response to your concerns. How does this change your review?

Please engage with the authors in the way that you would like reviewers to engage your submitted papers: critically and open to changing your mind. Thank you Reviewers jbMK and LsyV for your initial engagement!

Looking forward to the discussion!

---

### Meta-Review · Area_Chair_9kMc · 2022-08-26

**Recommendation:** Accept
**Confidence:** Certain

**Metareview:**

All reviewers agreed that this paper should be accepted because of the strong author response during the rebuttal phase. Specifically the reviewers appreciated the presentation of the paper, the inventive use of existing work, the simplicity and soundness of the method, and the strong theoretical guarantees and empirical results. Authors: please carefully revise the manuscript based on the suggestions by the reviewers: they made many careful suggestions to improve the work and stressed that the paper should only be accepted once these changes are implemented. Once these are done the paper will be a nice addition to the conference!

**Award:**

No

---

### Decision · Program_Chairs · 2022-09-14

Accept